# Deep Reinforcement learning on Adaptive Pairwise Critic and Asymptotic Actor

## Abstract

Maximum entropy deep reinforcement learning has displayed great potential on a range of challenging continuous tasks. The maximum entropy is able to encourage policy exploration, however, it has a tradeoff between the efficiency and stability, especially when employed on large-scale tasks with high state and action dimensionality. Sometimes the temperature hyperparameter of maximum entropy term is limited to remain stable at the cost of slower and lower convergence. Besides, the function approximation errors existing in actor-critic learning are known to induce estimation errors and suboptimal policies. In this paper, we propose an algorithm based on adaptive pairwise critics, and adaptive asymptotic maximum entropy combined. Specifically, we add a trainable state-dependent weight factor to build an adaptive pairwise target Q-value to serve as the surrogate policy objective. Then we adopt a state-dependent adaptive temperature to smooth the entropy policy exploration, which introduces an asymptotic maximum entropy. The adaptive pairwise critics can effectively improve the value estimation, preventing overestimation or underestimation errors. Meanwhile, the adaptive asymptotic entropy can adapt to the tradeoff between efficiency and stability, which provides more exploration and flexibility. We evaluate our method on a set of Gym tasks, and the results show that the proposed algorithms have better performance than several baselines on continuous control.

## 1 Introduction

The task of deep reinforcement learning (DRL) is to learn good policies by optimizing a discounted cumulative reward through function approximation. In DRL, the maximization over all noisy Q-value estimates at every update tends to prefer inaccurate value approximation that outweighes the true value Thrun & Schwartz (1993), i.e., the overestimation. This error further accumulates and broadcasts via bootstrapping of temporal difference learning Sutton & Barto (2018), which estimates the value function using the value estimate of a subsequent state. When function approximation is unavoidably adopted in the actor-critic setting on continuous control, the estimation errors are exaggerated. These errors may cause suboptimal policies, divergence and instability. To some extent, the inaccurate estimation is unavoidable in DRL because it is the basic trait for value-involved DRL to use random variables as target values. On the one hand, these stochastic target values will introduce some estimation biases. On the other hand, even an unbiased estimate with high variance can still lead to future overestimation in local regions of state space, which in turn can negatively affect the global policy Fujimoto et al. (2018). Therefore, diminishing the value variance without partiality can be an effective means to reduce estimation errors, no matter overestimation or underestimation. Taking the twin delayed deep deterministic policy gradient (TD3) Fujimoto et al. (2018) for example, always selecting the lower value from a pair of critics will induce an underestimation bias although it is beneficial for lower variance.

Several of recent works deal with errors like bootstrapping error caused by out-of-distribution (OOD) actions Kumar et al. (2019; 2020), and extrapolation error induced by the mismatch between the distribution of buffer-sampled data and true state-action visitation of the current policy Fujimoto et al. (2019). The authors in Wu et al. (2019) address the distribution errors by extra value penalty or policy regularization.

Overestimation is another induced errors, which was originally found in Q-learning algorithm by Watkins (1989), and was demonstrated in deep Q-network (DQN) Mnih et al. (2015) on discrete control. In recent years, overestimation is reported in function approximation of actor-critic methods on continuous control Fujimoto et al. (2018); Duan et al. (2021a). Although several algorithms are created to address the overestimation errors Fujimoto et al. (2018; 2019); Kumar et al. (2019); Wu et al. (2019); Duan et al. (2021a), the accuracy of function approximation is not flexibly touched since underestimation errors usually accompanies the correction to overestimation.

The paper has the following contributions. First, we propose the concept of adaptive pairwise critics, which connects a pair of critics using a trainable state-dependent weight factor, to combat estimation errors. Second, we propose the adaptive temperature which is also state-dependent so that the agent can freely explore with loose restriction on the selection of temperature hyperparameter. Based on this adaptive temperature, we organize a term of asymptotic maximum entropy to optimize the policy. The asymptotic maximum entropy is combined with the adaptive pairwise critics to serve the target Q-value as well as the surrogate policy objective. Third, we present a novel algorithm to tackle estimation errors and pursue effective and stable exploration. Finally, experimental evaluations are conducted to compare the proposed algorithm with several baselines in terms of sample complexity and stability.

## 2 RELATED WORK

In reinforcement learning, the agent needs to interact with the environment to collect enough knowledge for training. Without sufficient exploration, the collected data may be invalid for an optimal value. Therefore, reinforcement learning has to deal with the tradeoff between exploration and exploitation. There are several ways to enhance exploration in deep reinforcement learning (DRL), one of which is the off-policy approach which takes full advantage of past experience from replay buffer instead of on-policy data Mnih et al. (2015). Another method adopts policy exploration to stimulate the agent's motivation for a better balance between exploration and exploitation Mnih et al. (2016); Haarnoja et al. (2018a). Among them, soft actor-critic (SAC) Haarnoja et al. (2018a) achieves good performance on a set of continuous control tasks by adopting stochastic policies and maximum entropy. Stochastic policies generalize the policy improvement and introduce uncertainty into action decisions over deterministic counterparts Heess et al. (2015), and augmenting the reward return with an entropy maximization term encourages exploration, thus improving robustness and stability Ziebart et al. (2008); Ziebart (2010).

In recent years, many works have been proposed on top of SAC. The improvement of SAC can be realized by changing the rule of experience replay, for example, Wang & Ross (2019) samples more aggressively from recent experience while ordering the updates to ensure that updates from old data do not overwrite updates from new data, and Martin et al. (2021) relabels successful episodes as expert demonstrations for the agent to match. The distributional soft actor-critic (DSAC) Duan et al. (2021b); Ren et al. (2020); Ma et al. (2020); Duan et al. (2021c) combines the distributional return function within the maximum entropy to improve the estimation accuracy of the Q-value. It claims to prevent gradient explosion by truncating the difference between target and current return distributions, however, its assumptions of Gaussian distributions for random returns will induce more complexity and may not fit with the real distributions. Akimov (2019); Hou et al. (2020) reparameterize the reward representation and the policy, respectively, using a neural network transformation composed of multivariate factorization, and Ward et al. (2019) constructs normalizing flow policies before applying the squashing function to improving exploration within the SAC framework.

It is empirically shown that SAC is sensitive to the temperature hyperparameter. To provide flexibility for the choice of optimal temperature, SAC-v2 Haarnoja et al. (2018b) makes the first step to automatically tune the temperature hyperparameter by formulating a constrained optimization problem for the average entropy of policy. The dual to the constrained optimization will add an additional update procedure for the dual variable in determining the temperature. However, the assumption of convexity for theoretical convergence does not hold for neural networks, and the extra hyperparameter introduced by the transformation remains undetermined and needs more trials for generalization. Meta-SAC Wang & Ni (2020) uses metagradient along with a novel meta objective to automatically tune the entropy temperature in SAC. It distinguishes metaparameters from the learnable parameters and hyperparameters, and uses some initial states to train the meta temperature. However, due

to the limited data pool for the meta loss, the given experimental results show it does not perform better than SAC. Therefore, the auto adjustment of the temperature hyperparameter is still openly untouched for SAC.

The way to compute the target Q-value is an crucial design in DRL. The strategies include delayed update Van Hasselt et al. (2016), soft updates Lillicrap et al. (2015); Haarnoja et al. (2018b) and sophisticated ensembles Fujimoto et al. (2019); Kumar et al. (2019) of target Q-value. The sophisticated ensemble is some weighted mixture of the minimum and maximum among multiple learned Q-value functions, for example, TD3 adopts the minimum of pairwise critics, and bootstrapping error reduction (BEAR) Kumar et al. (2019) increases the number of Q-functions to 4. Behavior regularized actor critic (BRAC) Wu et al. (2019) investigated these design choices and concluded that the number of Q-functions over 2 only gives marginal improvement but significantly requires more computation cost. It is reported in Wu et al. (2019) that the minimum of two Q-functions adopted in TD3 outweighes a weighed mixture of Q-values in terms of simplicity and efficiency, however, there is a wide open unexplored area between them. How to design a mixture of Q-values is still largely left untouched.

## 3 BACKGROUND

We consider the infinite-horizon Markov Decision Process (MDP) in continuous action spaces, denoted by the tuple $(\mathcal{S}, \mathcal{A}, p, r)$ where $\mathcal{S}$ is the state space, $\mathcal{A}$ is the action space, $p(\cdot|s, a)$ is the transition probability of the next state $s' \in \mathcal{S}$ conditioned on the current state $s \in \mathcal{S}$ and action $a \in \mathcal{A}$, and $r \in \mathcal{S} \times \mathcal{A}$ is the reward which is the feedback from the environment of the current state $s$ and action $a$. The task of reinforcement learning (RL) is to learn an optimal policy that maximizes the reward return denoted by the expectation of discounted cumulative reward. DRL combines the neural networks with RL so that the reward return can be approximated by a parameterized function, where the agent follows a behavior policy $\pi$ to determine future rewards and next states. Let $p(\cdot|s, a)$ denotes the transition probability, then the surrogate function of the reward return can be selected as the action-value (Q-value) function with respect to the state-action pair in the form of

$$Q_\pi(s, a) = \mathbb{E}_{p^\pi(s_t|s_0, a_0)} \left[ \sum_{t=0}^{\infty} \gamma^t r(s_t, a_t) | s_0 = s, a_0 = a \right], \tag{1}$$

where $r(s, a)$ is the immediate reward produced by the state-action pair, and $\gamma \in (0, 1)$ is the discount horizon factor for future rewards. With the effect of behavior policy $\pi$, $p^\pi(s_t|s_0, a_0) = p(s_1|s_0, a_0) \prod_t \left[ \mathbb{E}_{a_{t-1} \sim \pi} p(s_t|s_{t-1}, a_{t-1}) \right]$ is the joint probability of all state-action pairs during an episode given the initial state-action pair $(s_0, a_0)$, and $\pi(a_{t+1}|s_{t+1})$ indicates the probability for the agent to choose the action $a_{t+1}$ given the state $s_{t+1}$.

Since the reward return has the property of Bellman equation, the temporal difference (TD) Tesauro (1995) is commonly used in the critic evaluation to minimize Bellman errors over sampled transitions $(s, a, r, s')$, which is given by $\mathbb{E}_{(s,a,r,s')} \left[ (r + \gamma Q^t(s', \pi(s')) - Q(s, a))^2 \right]$ Lillicrap et al. (2015), where $Q^t$ stands for the target Q-value. In off-policy algorithms using experience replay, $(s, a, r, s')$ is the tuple stored in the replay buffer at every environment step, $a$ is sampled from the experience pool, which is different from the on-policy next action $\pi(s')$. In the context, we use the term of 'iteration' to represent the index of updates. In the actor-critic paradigm, one iteration contains the evaluation step and the policy improvement step, which are used to update Q-value function and then optimize the policy. After the minimization of Bellman errors, the policy improvement is performed by maximizing the expected return $J(\theta) = \mathbb{E}_s [Q_\pi(s, \pi(s))]$. In some algorithms, the policy regularization may be attached to the expected return to smooth training Kumar et al. (2019); Jaques et al. (2019). These methods focus on constraining the policy gradient $\nabla_\theta J(\theta)$ to avoid gradient vanishing or exploding problems, which in turn reduces the estimation variance.

## 4 ADAPTIVE PAIRWISE CRITICS WITH ADAPTIVE ASYMPTOTIC ENTROPY

Value penalty or policy regularization is a common theme in DRL to improve stability, however, it tends to bring more hyperparameters for tuning, which will increase the difficulty for the designed algorithm to generalize to more tasks. Therefore, it is important for reasonable auto-adjustment for

these hyperparameters. The approaches to these adaptations are varied. For example, PPO adapts the penalty coefficient by setting some threshold values for the KL divergence, SAC-v2 Haarnoja et al. (2018b) automatically tune the temperature hyperparameter by adding a constraint solved by a related dual form, and Meta-SAC transforms the temperature hyperparameters into metaparameters.

Our work can be started by addressing a policy iteration method accompanying the adaptive pairwise critics and entropy estimation. We will first justify the adaptive pairwise critics and the adaptive asymptotic entropy, and verify the convergence of corresponding iterations, then organize the related algorithm with its usage of neural networks.

## 4.1 ADAPTIVE PAIRWISE CRITICS

The iteration of adaptive pairwise critic and adaptive asymptotic actor (APAA) is started by computing the revised target Q-value of a rollout following policy $\pi$, which is combined with a value penalty from the entropy exploration. Given the continuous MDP denoted by $(\mathcal{S}, \mathcal{A}, p, r)$, functions $Q_1, Q_2 : \mathcal{S} \times \mathcal{A} \to \mathbb{R}$ can be the Q-values of two critics, then a modified Bellman backup operator $\mathcal{T}^\pi$ is given by

$$\mathcal{T}^\pi Q(s_t, a_t) = r(s_t, a_t) + \gamma \mathbb{E}_{s_{t+1}, a_{t+1}} \left[ \overline{Q}(s_{t+1}, a_{t+1}) \right], \tag{2}$$

where $s_{t+1} \sim p(\cdot|s_t, a_t)$ and $a_{t+1} \sim \pi(\cdot|s_{t+1})$, and

$$\overline{Q}(s_t, a_t) = Q(s_t, a_t) - \alpha(\Lambda(s_t) + k_t) \log(\pi(a_t|s_t)) \tag{3}$$

is the APAA Q-value function, which can be obtained by repeatedly employing $\mathcal{T}^\pi$ for any policy $\pi$. $\Lambda(s_t)$ is the adaptive random variable (ARV) dependent on the state $s_t$, $0 \leq k_t \leq 1$ is the asymptotic variable gradually increasing from 0 to 1 as the time step proceeds, and $\alpha$ is the fixed temperature hyperparameter. The sum of ARV and the asymptotic variable compose the adaptive asymptotic temperature for the entropy. The joint Q-value function $Q$ is formularized as

$$Q(s_t, a_t) = (1 - \Gamma(s_t))Q_1(s_t, a_t) + \Gamma(s_t)Q_2(s_t, a_t), \tag{4}$$

where $0 \leq \Gamma(\cdot) \leq 1$ is the state-dependent adaptive random weight (ARW) to adjust the influence of two critics.

**Lemma 1.** *Consider the sequence $Q_{k+1} = \mathcal{T}^\pi Q_k$, then given the condition that the Q-values are bounded, i.e., $|Q_1(s,a)| < \infty$, $|Q_2(s,a)| < \infty$, $\forall(s,a) \in \mathcal{S} \times \mathcal{A}$, the sequence $Q_k$ will converge to a unique optimal value as $k \to \infty$.*

The proof of Lemma 3 can be found in Appendix A (in Supplementary Files), however, the sufficient condition is not always satisfied when the function approximation is applied. Since the state-action spaces are continuous and the transition probability is unknown in model-free DRL, the Q-value function cannot be formulated or tabulated by the state-action pairs, which means the function approximation gives no absolute guarantee for the bounded Q-values. Therefore, instead of repeatedly applying (2) directly by equality, the practical evaluation step is estimated by minimizing the expected mean square error (MSE) between $\mathcal{T}^\pi \overline{Q}_k(s, a)$ and $Q_{1,k+1}(s, a)$ or $Q_{2,k+1}(s, a)$. Once the expected MSE converges to zero, the two Q-value functions updated based on (2) will end up with little fluctuation around the same fixed point when the hyperparameters are chosen properly.

## 5 ADAPTIVE ASYMPTOTIC ENTROPY

When it comes to the policy improvement step, the purpose of APAA iteration contains two points, which aim to improve the Q-value for each update as while as projecting the policy onto a normalized distribution. The policy update step is given by

$$\begin{aligned} &\pi_{new} \\ &= \arg \max_{\pi \in \Pi} \mathbb{E}_{s_t, a_t} \left[ Q(s_t, a_t) - \alpha(\Lambda(s_t) + k_t) \log(\pi(a_t|s_t)) \right] \\ &= \arg \min_{\pi \in \Pi} \mathbb{E}_{s_t} \left[ D_{KL} \left( \pi(\cdot|s_t) \| \exp \left( \frac{Q(s_t, \cdot)}{\alpha\Lambda(s_t) + \alpha k_t} \right) \right) \right], \end{aligned} \tag{5}$$

where $s_t \sim \mathcal{S}, a_t \sim \pi(\cdot|s_t)$, and the choice of policy $\pi$ is limited to a set of parameterized Gaussian distributions $\prod$ for flexibility. With modest computation, the second equality holds, which can be found in Appendix B. The KL divergence $D_{KL}$ shows that the improved policy is updated towards the distribution constituted by the exponential of the normalized Q-value function. The adaptive asymptotic temperature $\Lambda(s_t) + k_t$ is not dependent on the action, and thus does not contribute to the policy gradient. However, it is still important to be chosen for a better insurance of the expected policy improvement.

In continuous control problems of model-free DRL, where the transition probability is unknown and the state and action spaces are both continuous, it is not possible to provides policy improvement at every state-action point over $\mathcal{S} \times \mathcal{A}$, which we call as the absolute policy improvement in this paper. Therefore, we propose a practical standard for the policy improvement, which is named as the expected policy improvement. It shows that the projected policy in (17) can produce higher updated Q-value with expression given by (1), and the result is organized in Lemma 4.

**Lemma 2.** *Denote $\pi_{new}$ and $\pi_{old}$ as the policies before and after the update defined in* (17)*, respectively. Then the expected policy improvement, i.e., $\mathbb{E}_{(s_t,a_t) \sim \mathcal{S} \times \mathcal{A}}[Q_{\pi_{new}}(s_t, a_t) - Q_{\pi_{old}}(s_t, a_t)] \geq 0$, can be guaranteed.*

The proof of Lemma 4 can be found in Appendix C. Besides projecting the policy into a selected set of distributions, (17) also maximizes the expectation of APAA Q-value function defined in (3) by choosing the specific adaptive asymptotic temperature $\Lambda(s_t) + k_t$, which is the key to the guarantee of the wanted expected policy improvement, shown in the second step of proving Lemma 4. Furthermore, in discrete control problems, where the state-action spaces are both discrete and bounded, the absolute policy improvement can be realized by removing the expectation over $s \sim \mathcal{S}$ in (17).

The APAA iteration alternates between the policy evaluation and the expected policy improvement steps, and will converge to the optimal policy which provides higher expected Q-value than the other policies in $\prod$. The theorem describing the APAA iteration is organized in

**Theorem 1.** *Let $l_t$ be the learning rate at time step $t$, then given the condition that*

$$0 \leq l_t(x) \leq 1, \sum_t l_t(x) = \infty, \sum_t l_t^2(x) < \infty \ w.p.1., \tag{6}$$

*repeated application of policy evaluation and expected policy improvement will converge to an optimal policy $\pi^\star \in \prod$ such that $\mathbb{E}_{(s_t,a_t) \sim \mathcal{S} \times \mathcal{A}}[Q_{\pi^\star}(s_t, a_t) - Q_\pi(s_t, a_t)] \geq 0, \forall \pi \in \prod$.*

*Proof*    See Appendix D.                                                                                      □

## 5.1 Algorithm of Adaptive Pairwise Critics with Adaptive Asymptotic Entropy

We have discussed above the practical scenario of Theorem 2 in large continuous domains, which requires parameterized function approximations for both the Q-value function and the policy. To stabilize the training process, separated current and target networks are provided for both the Q-value function and the policy. Based on these parameterized networks and (2), the loss function for the update of critic parameters in the policy evaluation step can be estimated by

$$L(\omega^i) = \mathbb{E}_{(s,a,r,s')}\left[\frac{1}{2}(r + \gamma Q^t(s', a') - Q(s, a))^2\right], \tag{7}$$

where $a' = \pi_{\theta'}(s')$ is the action following the target policy parameterized by $\theta'$, and $(s, a, r, s')$ is a tuple of history data sampled from the experience pool. And

$$Q^t(s', a') = (1 - \Gamma_{\mu'}(s'))Q_{\omega'^1}(s', a') + \Gamma_{\mu'}(s')Q_{\omega'^2}(s', a')$$
$$- \alpha(\Lambda_{\lambda'}(s') + k)\log(\pi_{\theta'}(a'|s')), \tag{8}$$

$$Q(s, a) = (1 - \Gamma_\mu(s))Q_{\omega^1}(s, a) + \Gamma_\mu(s)Q_{\omega^2}(s, a), \tag{9}$$

where $\omega^1, \omega^2, \omega'^1$ and $\omega'^2$ parameterize two critic networks and their target estimates, respectively. Besides, $\Lambda_{\lambda'}$ is target ARV parameterized by $\lambda'$, $k$ is the asymptotic variable increasing from 0 to 1 as the time step proceeds, the state-dependent ARW $\Gamma$ parameterized by $\mu$ and $\mu'$ is clipped in $[0, 1]$ to determine the influence of two Q-value functions, and $\pi_{\theta'}(\cdot|s')$ is the target policy distribution

conditioned on the next state $s'$. By minimizing (7), the critic parameters can be updated for each policy evaluation step. Then (7) can be optimized with stochastic gradient

$$\hat{\nabla}_{\omega^i} L(\omega^i) = \mathbb{E}_{(s,a,r,s')}[\hat{\nabla}_{\omega^i} Q(s,a)(Q(s,a) - r - \gamma Q^t(s',a'))] \quad for \quad i \in \{1,2\}. \tag{10}$$

It is noticeable that two extra networks has been added for ARV and ARW, however, to avoid introducing extra more saddle point problems, we see their trainable parameters as part of the actor parameter, i.e., the actor parameter is composed of the policy parameter, the ARV parameter and the ARW parameter. Then the surrogate objective function to update the current actor parameter $(\theta, \lambda, \mu)$ in the expected policy improvement step (see Lemma 4) can be given by

$$J(\theta, \lambda, \mu) = \mathbb{E}_s \left[ Q(s,a) - \alpha(\Lambda_\lambda(s) + k) \log(\pi_\theta(a|s)) \right], \tag{11}$$

where $s$ comes from the tuple of history data, $a = \pi_\theta(s)$ is the reparameterized action based on $s$ and the policy network parameterized by $\theta$. $\Lambda_\lambda$ is current ARV parameterized by $\lambda$, $\Gamma_\mu$ is current ARW parameterized by $\mu$, and $\pi_\theta(\cdot|s)$ is the current policy distribution conditioned on the current state $s$. By maximizing (11), the actor parameter can be updated for policy improvement each step. The gradient of (11) is computed as

$$\hat{\nabla}_\theta J(\theta, \lambda, \mu) = \mathbb{E}_s \left[ \hat{\nabla}_a Q(s,a) \hat{\nabla}_\theta \pi_\theta(s) - \frac{\alpha \hat{\nabla}_\theta \pi_\theta(a|s)(\Lambda_\lambda(s) + k)}{\pi_\theta(a|s)} \right], \tag{12}$$

$$\hat{\nabla}_\lambda J(\theta, \lambda, \mu) = \mathbb{E}_s \left[ -\alpha \hat{\nabla}_\lambda \Lambda_\lambda(s) \log(\pi_\theta(a|s)) \right], \tag{13}$$

$$\hat{\nabla}_\mu J(\theta, \lambda, \mu) = \mathbb{E}_s \left[ \hat{\nabla}_\mu Q(s,a) \right]. \tag{14}$$

Then the target parameters $(\omega'^1, \omega'^2, \theta', \lambda', \mu')$ are updated following the "soft" target updates Lillicrap et al. (2015) by $(\omega^1, \omega^2, \theta, \lambda, \mu)$, in the way of

$$\begin{aligned}
\omega'^i_{t+1} &\leftarrow \tau \omega^i_{t+1} + (1-\tau)\omega'^i_t \quad for \quad i \in \{1,2\} \\
\theta'_{t+1} &\leftarrow \tau \theta_{t+1} + (1-\tau_1)\theta'_t, \\
\lambda'_{t+1} &\leftarrow \tau \lambda_{t+1} + (1-\tau_1)\lambda'_t, \\
\mu'_{t+1} &\leftarrow \tau \mu_{t+1} + (1-\tau_1)\mu'_t,
\end{aligned} \tag{15}$$

where $0 \leq \tau < 1$ is the factor to control the speed of policy updates for the sake of small value error at each iteration, and $0 \leq \tau_1 < 1$ is set as 1 in our application. We organize the above procedures as the adaptive pairwise critics with adaptive asymptotic entropy (APAA) algorithm, whose pseudocode is described by Algorithm 1. The algorithm alternates between running the environment steps to collect experience and updating the network parameters using the stochastic gradients computed by the sampled batches from the experience pool. In (10), (12), (13) and (14), the gradients are in their expectation forms, however, practically they are averaged over the results of sampled tuples, which usually follow policies parameterized by different parameters in off-policy methods. In some algorithms, one gradient step follows one or several environment steps to stabilize the training process.

# 6 EXPERIMENTS

## 6.1 BENCHMARKS

The performance of our proposed method is compared with several prior model-free reinforcement learning algorithms in terms of the sample complexity and stability on a set of gym continuous control tasks from the MuJoCo suite Todorov et al. (2012); Brockman et al. (2016). Fig. 1 shows the illustrations of benchmarks adopted in this paper.

## 6.2 BASELINES

The adopted baselines include deep deterministic policy gradient (DDPG) Lillicrap et al. (2015), TD3, SAC and BRAC. Before the existence of SAC, DDPG is regarded as one of the most efficient

---

**Algorithm 1** APAA Algorithm

---

1: **Input**: The update maximum time step $T$
2: Initialize parameters $\omega^1 \leftarrow \omega_0^1, \omega^2 \leftarrow \omega_0^2, \theta \leftarrow \theta_0, \lambda \leftarrow \lambda_0, \mu \leftarrow \mu_0$
3: Initialize target parameters $\omega'^1 \leftarrow \omega_0'^1, \omega'^2 \leftarrow \omega_0'^2, \theta' \leftarrow \theta_0', \lambda' \leftarrow \lambda_0', \mu' \leftarrow \mu_0'$
4: Initialize the learning rates $l_c, l_a$ for the critic and the actor, the time step $t \leftarrow 0$, the soft update
   hyperparameter $\tau$, the maximum time step $T$, the batch size $B$ and the replay buffer $\mathcal{D} \leftarrow \emptyset$.
5: **while** $t < T$ **do**
6:     Select action $a_t \sim \pi_{\theta_t}(a_t|s_t)$
7:     Observe the reward and next state $s_{t+1}, r_t \sim p(s_{t+1}|s_t, a_t)$
8:     Store transition $\mathcal{D} \leftarrow \mathcal{D} \cup \{(s_t, a_t, r_t, s_{t+1})\}$
9:     Sample a batch of transitions $\mathcal{B} = (s, a, r, s')_{i=1}^B$ from $\mathcal{D}$
10:     **for** each time step **do**
11:         $\omega_{t+1}^i \leftarrow \omega_t^i - l_c \hat{\nabla}_{\omega_t^i} L(\omega_t^i)$ for $i \in \{1, 2\}$ following Eq. (10)
12:         $\theta_{t+1} \leftarrow \theta_t + l_a \hat{\nabla}_{\theta_t} J(\theta_t, \lambda_t, \mu_t)$ following Eq. (12)
13:         $\lambda_{t+1} \leftarrow \lambda_t + l_a \hat{\nabla}_{\lambda_t} J(\theta_t, \lambda_t, \mu_t)$ following Eq. (13)
14:         $\mu_{t+1} \leftarrow \mu_t + l_a \hat{\nabla}_{\mu_t} J(\theta_t, \lambda_t, \mu_t)$ following Eq. (14)
15:         $\omega'^i_{t+1} \leftarrow \tau \omega_{t+1}^i + (1 - \tau) \omega'^i_t$ for $i \in \{1, 2\}$ following Eq. (15)
16:     **end for**
17:     $s_{t+1} \leftarrow s_t$
18:     $t \leftarrow t + 1$
19: **end while**

---

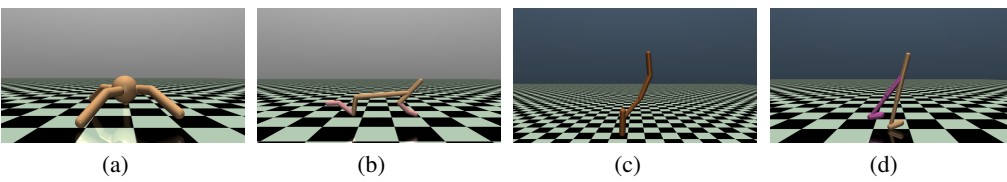

Figure 1: (a) Ant-v3; (b) Halfcheetah-v3; (c) Hopper-v3; (d) Walker2d-v3

off-policy DRL methods Duan et al. (2016), followed by TD3 as an extension. SAC has achieved model-free state-of-the-art sample efficiency in multiple challenging continuous control domains Christodoulou (2019), and BRAC is can be seen as a variant of SAC by adopting an extra policy regularization based on the KL divergence between policies before and after updating.

Our proposed algorithm shares the same set of hyperparameters with other baselines to keep fairness. The gaussian exploration noise with a fixed variance of 0.2 is added to the action at every time step, then the noisy action is clipped within the set boundary. With the discount horizon factor chosen as 0.99, algorithms including the proposed one, SAC and BRAC adopt the entropy term, which is computed by normal random policies, whose mean and variance are parameterized by fully connected networks with two hidden layers, each of which has 256 units. Except that, both DDPG and TD3 use deterministic policies, also parameterized by fully connected networks with two hidden layers. We organize the network architectures and hyperparameters in Appendix E and F, respectively. The Adam optimizer Kingma & Ba (2014) is used to update the network parameters.

### 6.3 RESULTS

We train 10 seeds for each algorithm to keep a fair comparison. After every 500 iterations (time steps), we launch a evaluation procedure, which averages 10 rollouts for a test. The average reward of a test will be recorded at every evaluation procedure, and all tests throughout the time step scale give the result of each algorithm.

The average rewards of algorithms tested in chosen benchmarks are shown Fig. 2 with 95% confidence interval (CI). From Figs. 5(a), 5(b) and 5(c), we can observe overwhelmed advantage of APAA over other baselines. In Hopper environment, since the converged value is far lower than other benchmarks, the tolerance for the fluctuation around convergence is much lower, which causes

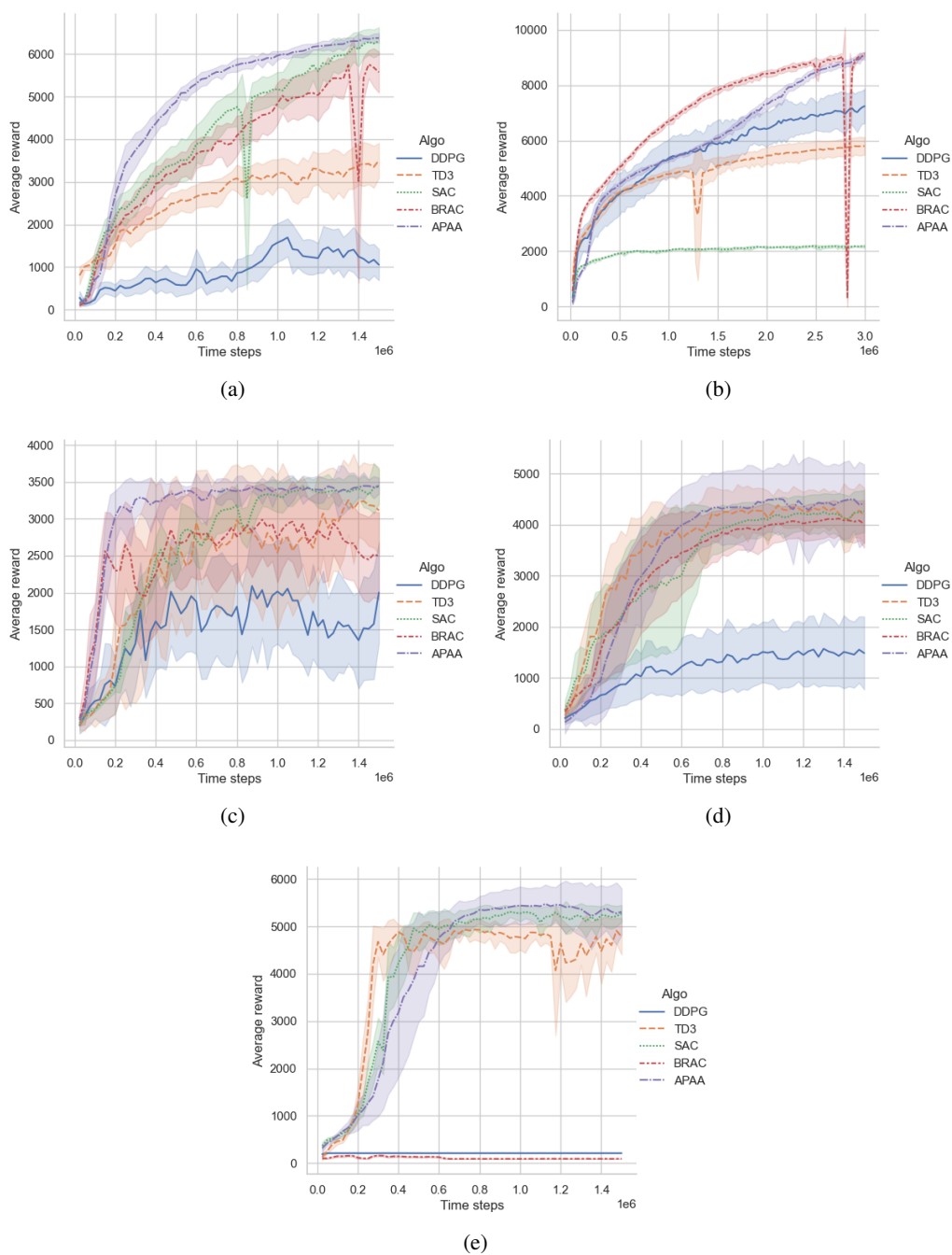

Figure 2: Average reward versus time step in (a) Ant-v3; (b) Halfcheetah-v3; (c) Hopper-v3; (d) Walker2d-v3; (e) Humanoid-v3

the instability problems of tested baselines. However, APAA shows strong robustness and has a converged value up to 3500 compared with other baselines, as shown in 5(c). In Fig. 5(d), APAA still shows better performance than that of other baselines, and converges around 5000. For Humanoid with high-dimensional action space, Fig. 5(e) shows that APAA is much better than other baselines and can steadily converge to 6000. Over all figures in Fig. 2, SAC and BRAC both have their up and downs, and DDPG gives the worst performance, considering the fact that the double critics are not employed in DDPG to reduce potential overestimation. Due to the stochastic property of random

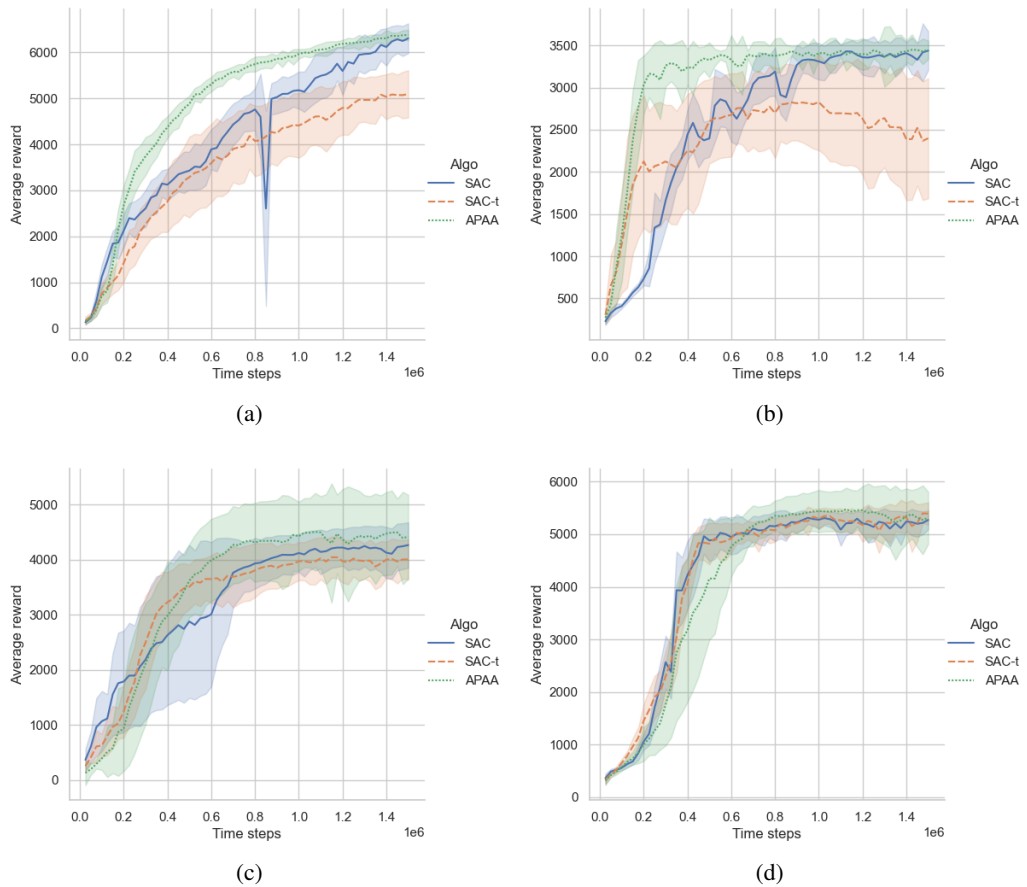

Figure 3: Comparison between APAA and automating entropy adjustment of SAC in (a) Ant-v3; (b) Hopper-v3; (c) Walker2d-v3; (d) Humanoid-v3

variables, the best performance cannot be ensured for every seed, which means the potential reduced stability and convergence (partly told by Fig. 2) are reasonable.

Since SAC has an variant working on automatic adjustment of the temperature hyperparameter Haarnoja et al. (2018b), we use SAC-t to represent it and compare its performance with APAA in Fig. 3 with 95% CI. SAC-t adds an extra hyperparameter $\overline{\mathcal{H}}$ as the target entropy in exchange of the temperature, which may not lead to better performance because the target entropy cannot be generalized and also needs automatic tuning, as reported by Wu et al. (2019). According to Fig. 3, SAC-t fails to produce better performance than APAA given the choice of the target entropy as 0.5, which implies the right way of adjusting the temperature in APAA.

Due to the page limit, we make the comparison of value estimates in Appendix G.

## 7 CONCLUSION

In this paper, we proposed a state-dependent adaptive temperature to encourage policy exploration, which can strike a better balance between the efficiency and stability by introducing an asymptotic maximum entropy. Then the asymptotic maximum entropy is combined with the adaptive pairwise critics to benefit the policy evaluation and improvement steps. Based on the above two components, we present APAA to gain better tradeoff between efficiency and stability. We evaluate our method on a set of Gym tasks, and the results show that the proposed algorithms have better performance than several baselines on continuous control.

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

## A    PROOF OF LEMMA 3

**Lemma 3.** *Consider the sequence $Q_{k+1} = \mathcal{T}^\pi Q_k$, then given the condition that the Q-values are bounded, i.e., $|Q_1(s,a)| < \infty$, $|Q_2(s,a)| < \infty$, $\forall(s,a) \in \mathcal{S} \times \mathcal{A}$, the sequence $Q_k$ will converge to a unique optimal value as $k \to \infty$.*

*Proof*

$$
|\mathcal{T}^\pi Q(s_t, a_t) - \mathcal{T}^\pi Q'(s_t, a_t)|
$$
$$
= \gamma \left| \mathbb{E}_{s_{t+1} \sim p, a_{t+1} \sim \pi} \left[ Q(s_{t+1}, a_{t+1}) - Q'(s_{t+1}, a_{t+1}) \right] \right|
$$
$$
\leq \gamma \mathbb{E}_{s_{t+1} \sim p, a_{t+1} \sim \pi} \left[ |Q(s_{t+1}, a_{t+1}) - Q'(s_{t+1}, a_{t+1})| \right]
$$
$$
\leq \gamma \max_{s_{t+1} \sim p, a_{t+1} \sim \pi} |Q(s_{t+1}, a_{t+1}) - Q'(s_{t+1}, a_{t+1})|
$$
$$
= \gamma \|Q - Q'\|_\infty, \tag{16}
$$

where $\| \cdot \|_\infty$ means the max norm, $p$ and $\pi$ is short for $p(\cdot|s_t, a_t)$ and $\pi(\cdot|s_t)$. Since the Q-values are assumed to be bounded, the adaptive pairwise critics $Q$ is also bounded, then the third inequality holds. We reach a conclusion that $\forall(s_t, a_t) \in \mathcal{S} \times \mathcal{A}$, (16) holds, which can be rewritten as max-norm contraction mapping as $\|T^\pi Q - T^\pi Q'\|_\infty \leq \gamma \|Q - Q'\|_\infty$. According to the property of contraction operator, the sequence $Q_{k+1} = \mathcal{T}^\pi Q_k$ will converge to its fixed point. $\square$

## B    PROOF OF (17)

$$
\pi_{new}
$$
$$
= \arg \max_{\pi \in \Pi} \mathbb{E}_{s_t, a_t} \left[ Q(s_t, a_t) - \alpha(\Lambda(s_t) + k_t) \log(\pi(a_t|s_t)) \right]
$$
$$
= \arg \min_{\pi \in \Pi} \mathbb{E}_{s_t} \left[ D_{KL} \left( \pi(\cdot|s_t) \| \exp \left( \frac{Q(s_t, \cdot)}{\alpha \Lambda(s_t) + \alpha k_t} \right) \right) \right], \tag{17}
$$

*Proof*

$$
\pi_{new} = \arg \max_{\pi \in \Pi} \mathbb{E}_{s_t \sim \mathcal{S}, a_t \sim \pi(\cdot|s_t)} \left[ Q(s_t, a_t) - \alpha(\Lambda(s_t) + k_t) \log(\pi(a_t|s_t)) \right]
$$
$$
= \arg \min_{\pi \in \Pi} \mathbb{E}_{s_t \sim \mathcal{S}, a_t \sim \pi(\cdot|s_t)} \left[ \log(\pi(a_t|s_t)) - \frac{Q(s_t, a_t)}{\alpha(\Lambda(s_t) + k_t)} \right]
$$
$$
= \arg \min_{\pi \in \Pi} \mathbb{E}_{s_t \sim \mathcal{S}, a_t \sim \mathcal{A}} \left[ \pi(a_t|s_t) \left[ \log(\pi(a_t|s_t)) - \frac{Q(s_t, a_t)}{\alpha(\Lambda(s_t) + k_t)} \right] \right]
$$
$$
= \arg \min_{\pi \in \Pi} \mathbb{E}_{s_t \sim \mathcal{S}} \left[ D_{KL} \left( \pi(\cdot|s_t) \| \exp \left( \frac{Q(s_t, \cdot)}{\alpha(\Lambda(s_t) + k_t)} \right) \right) \right], \tag{18}
$$

where the second equality holds because $\Lambda(s_t)$ and $k_t$ are not dependent on $a_t$, and $D_{KL}(\cdot\|\cdot)$ is the KL divergence. $\square$

## C    PROOF OF LEMMA 4

**Lemma 4.** *Denote $\pi_{new}$ and $\pi_{old}$ as the policies before and after the update defined in (17), respectively. Then the expected policy improvement, i.e., $\mathbb{E}_{(s_t, a_t) \sim \mathcal{S} \times \mathcal{A}}[Q_{\pi_{new}}(s_t, a_t) - Q_{\pi_{old}}(s_t, a_t)] \geq 0$, can be guaranteed.*

*Proof*

$$
var(\mathcal{T}^{\pi_{new}} Q_{\pi_{old}} - Q_{\pi_{new}})
$$
$$
= \mathbb{E} \left[ (\mathcal{T}^{\pi_{new}} Q_{\pi_{old}} - Q_{\pi_{new}})^2 \right] - (\mathbb{E}[\mathcal{T}^{\pi_{new}} Q_{\pi_{old}} - Q_{\pi_{new}}])^2
$$
$$
\geq 0, \tag{19}
$$

where $var(\cdot)$ represents the variance. According to (19), we have

$$(\mathbb{E}[\mathcal{T}^{\pi_{new}}Q_{\pi_{old}} - Q_{\pi_{new}}])^2 \leq \mathbb{E}\left[(\mathcal{T}^{\pi_{new}}Q_{\pi_{old}} - Q_{\pi_{new}})^2\right], \tag{20}$$

then $\mathbb{E}[\mathcal{T}^{\pi_{new}}Q_{\pi_{old}}]$ will converge to $\mathbb{E}[Q_{\pi_{new}}]$ based on the expected MSE analyzed in the paragraph after Lemma 3. This constitutes the first step of the proof for the expected policy improvement, which can be written as

$$
\begin{aligned}
&\mathbb{E}_{(s_t,a_t)\sim\mathcal{S}\times\mathcal{A}}[Q_{\pi_{new}}(s_t, a_t)] \\
&= \mathbb{E}_{(s_t,a_t)\sim\mathcal{S}\times\mathcal{A}}[\mathcal{T}^{\pi_{new}}Q_{\pi_{old}}(s_t, a_t)] \\
&\geq \mathbb{E}_{(s_t,a_t)\sim\mathcal{S}\times\mathcal{A}}[\mathcal{T}^{\pi_{old}}Q_{\pi_{old}}(s_t, a_t)] \\
&= \mathbb{E}_{(s_t,a_t)\sim\mathcal{S}\times\mathcal{A}}[r(s_t, a_t)] + \gamma\mathbb{E}_{s_{t+1}\sim p(\cdot|s_t,a_t),a_{t+1}\sim\pi(\cdot|s_{t+1})}[\overline{Q}_{\pi_{old}}(s_{t+1}, a_{t+1})] \\
&\geq \mathbb{E}_{(s_t,a_t)\sim\mathcal{S}\times\mathcal{A}}[r(s_t, a_t)] + \gamma\mathbb{E}_{s_{t+1}\sim p(\cdot|s_t,a_t),a_{t+1}\sim\pi(\cdot|s_{t+1})}[Q_{\pi_{old}}(s_{t+1}, a_{t+1})] \\
&= \mathbb{E}_{(s_t,a_t)\sim\mathcal{S}\times\mathcal{A}}[Q_{\pi_{old}}(s_t, a_t)],
\end{aligned}
\tag{21}
$$

where the second inequality holds because of the update rule following the first equality of (17), the third equality is the expected form of modified Bellman backup operator, the forth inequality holds because both the entropy and the adaptive asymptotic temperature are nonnegative, and the last equality is a variant of the bellman equation. Because of the unknown transition probability and continuous state-action spaces in continuous model-free DRL, the Q-value function is usually approximated by neural networks, which makes it impossible to directly apply the bellman equation to every state-action pair over $\mathcal{S} \times \mathcal{A}$. Under the circumstance, the bellman equation only holds in statistical sense. $\square$

## D    PROOF OF THEOREM 2

**Theorem 2.** *Let $l_t$ be the learning rate at time step $t$, then given the condition that*

$$0 \leq l_t(x) \leq 1, \sum_t l_t(x) = \infty, \sum_t l_t^2(x) < \infty \ w.p.1., \tag{22}$$

*repeated application of policy evaluation and expected policy improvement will converge to an optimal policy $\pi^\star \in \prod$ such that $\mathbb{E}_{(s_t,a_t)\sim\mathcal{S}\times\mathcal{A}}[Q_{\pi^\star}(s_t, a_t) - Q_\pi(s_t, a_t)] \geq 0, \forall\pi \in \prod$.*

*Proof*    According to Lemma 1 of (SINGH et al., 2000), the condition (22) can make the expected MSE of temporal difference (TD) converge to zero, which validates the policy evaluation of adaptive asymptotic iteration and prove the first step of (21) to be true. With the monotonic increasing of the updated expected Q-value, the converged optimal policy will render $\mathbb{E}_{(s_t,a_t)\sim\mathcal{S}\times\mathcal{A}}[Q_{\pi^\star}(s_t, a_t) - Q_\pi(s_t, a_t)] \geq 0, \forall\pi \in \prod$. $\square$

## E    NETWORK ARCHITECTURE

We construct the critic network using a fully-connected MLP with two hidden layers. The input is composed of the state and action, outputting a value representing the Q-value. The ReLU functions are adopted to activate the two hidden layers. The setting of policy network follows normal random distribution, whose expectation and variance are fully-connected networks fed only by the state. Both of them have two hidden layers activated by the ReLU function. After the hidden layers, a Tanh function and a Softplus function follows to form the expectation and variance, respectively. With the expectation and variance, a normal distribution can be achieved to represent the random policy. The network of state-dependent ARV $\Lambda$ and ARW $\Gamma$ are constructed similar to either the expectation and variance of policy network except replacing the last nonlinearity activation by a Sigmoid function. The architecture of networks are plotted in Fig. 4.

The above mentioned network architecture is adopted for the random policy. For the algorithm using the deterministic policy, the critic is constructed in the same way, however, the actor network is the same as that of the expectation of normal random distribution.

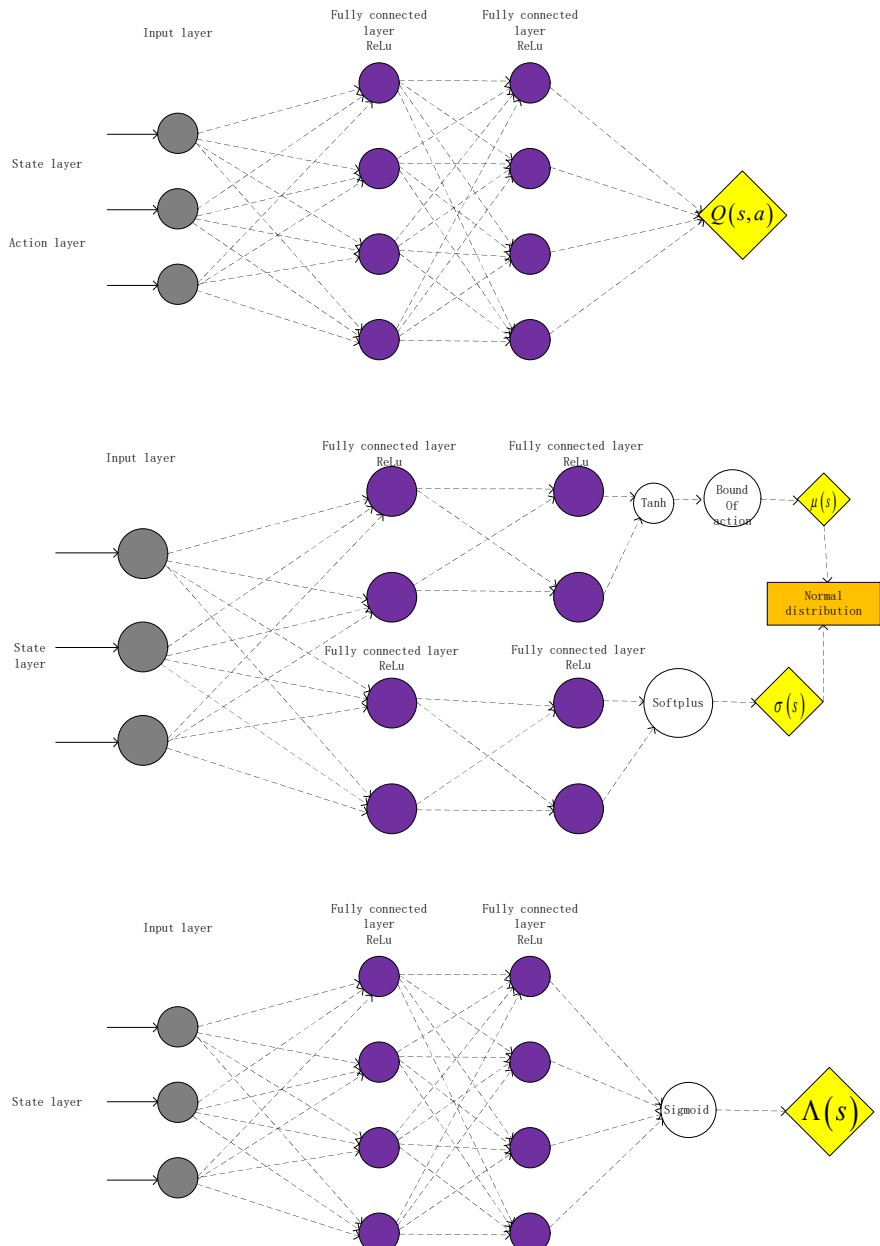

Figure 4: Architecture of networks.

## F HYPERPARAMETERS

Table 1 lists the common hyperparameters shared by all experiments and their respective settings. In this table, $LR\_a$ means the learning rate of the actor (includes $lambda$ in our proposed algorithm), and $LR\_c$ means the learning rate of critics. $\tau\_a$ and $\tau\_c$ represent soft update hyperparameter of the actor and the critic, respectively, and $\tau\_a = 1$ means we adopt immediate update for the actor. The symbol $var$ represents the variance of gaussian exploration noise, and $\alpha$ is the fixed temperature hyperparameter, which is applied in algorithms except DDPG and TD3. $\alpha_d$ represents the Wight factor of KL divergence for policy regularization applied in BRAC, and $\beta$ is the asymptotic rise

Table 1: **List of hyperparameters**

| Hyperparameter | Value | Description | Algorithm applied |
|:---:|:---:|:---:|:---:|
| $LR\_a$ | 0.0003 | Learning rate of actor | All |
| $LR\_c$ | 0.0003 | Learning rate of critic | All |
| $\tau\_a$ | 1 | Soft update parameter of actor | All |
| $\tau\_c$ | 0.005 | Soft update parameter of critic | All |
| $\gamma$ | 0.99 | Discount horizon factor | All |
| $var$ | 0.2 | The variance of exploration noise | All |
| $\alpha$ | 0.1 | Fixed temperature | Except DDPG and TD3 |
| $\alpha_d$ | 0.1 | Wight factor of KL regularization | BRAC |
| $\beta$ | 0.9995 | Asymptotic rise rate | APAA |
| $\overline{\mathcal{H}}$ | 0.5 | Target entropy | SAC-t |
| $Batch$ | 256 | Size of each mini-batch | All |
| $Units$ | 256 | Hidden layer units | All |
| $Memory$ | 1000000 | Size of replay buffer | All |
| $Interval$ | 500 | Evaluation period | All |
| $Test$ | 10 | Rollouts per evaluation | All |

rate for $k_t = 1 - \beta^t$. Besides, we choose the target entropy $\overline{\mathcal{H}}$ as $0.5$ for the automating entropy adjustment of SAC (SAC-t).

Moreover, $Batch$ represents the size of mini-batches sampled for training, and $Memory$ is short for the size of replay buffer. The rest in Table 1 are the hyperparameters for the evaluation procedure, specifically, $Interval$ means how many time steps between two successive evaluation procedures, and $Test$ means the number of rollouts run during each evaluation procedure.

# G VALUE ESTIMATE

We plot the value estimate, approximated by the trained Q-value networks, over time steps to compare with the true value, which is represented by the discount return of a rollout starting from $1000$ random state-action pairs from the replay buffer. The discount return of a rollout is recorded every $500$ time steps, which follows the updated policy at that time step and is different from the average return. The differences between the value estimates and the true values is illustrated in Fig. 5. From these figures, we can observe that the algorithm without tuning the target Q-value (DDPG) suffers great overestimation, however, simply choosing the smaller Q-value from a pair of critics (TD3 and SAC) will bring nonnegligible underestimation, instead. Since inaccurate value estimates will lead to poor policy updates, neither underestimation or overestimation is wanted. Dynamic adjustment of target Q-value used in ARW of APAA provides a preference.

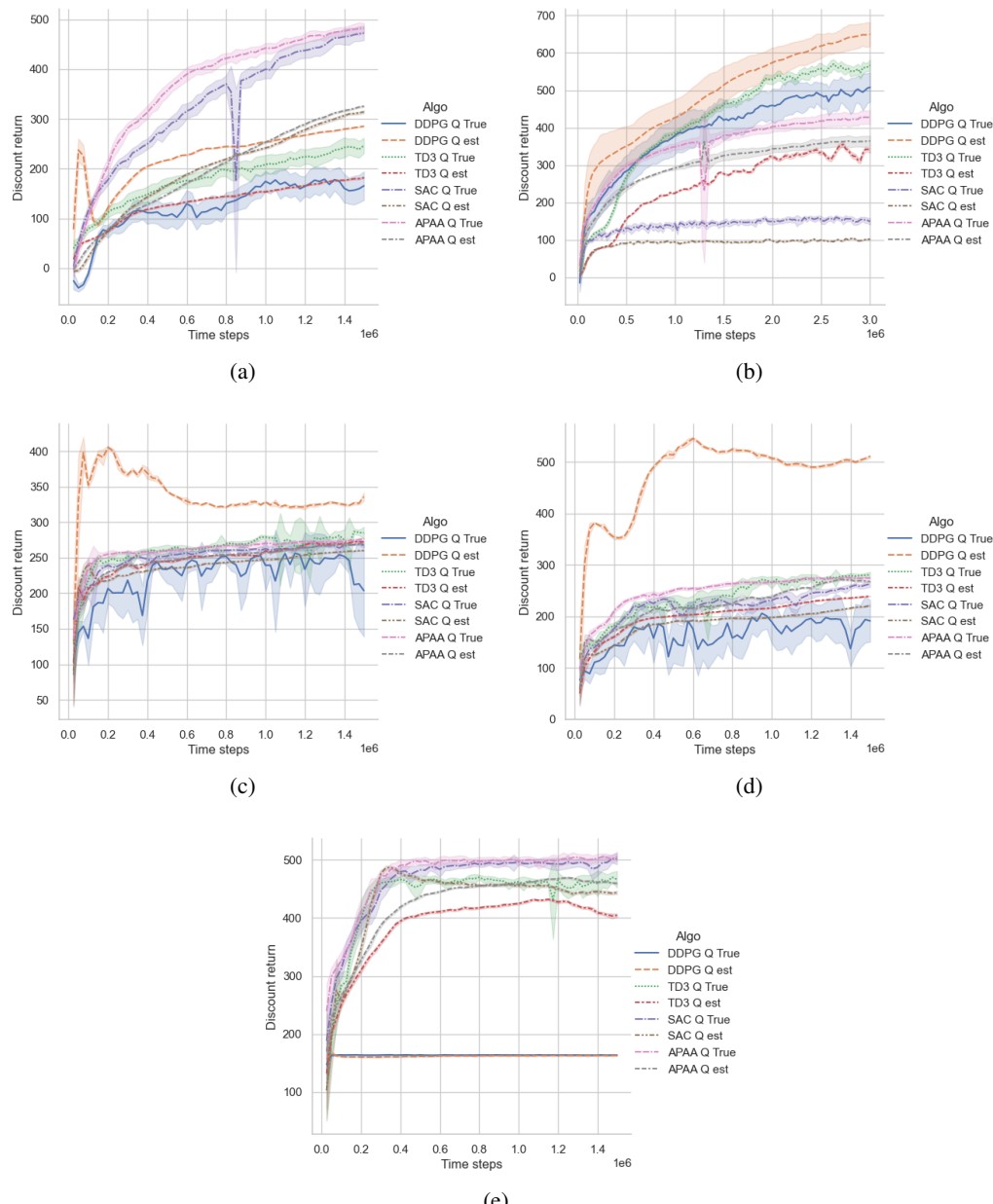

Figure 5: Comparison between the value estimate and the true value in (a) Ant-v3; (b) Halfcheetah-v3; (c) Hopper-v3; (d) Walker2d-v3; (e) Humanoid-v3. 'Q True' means the true value and 'Q est' means the value estimate.

