# OpenReview forum: "Deep Reinforcement learning on Adaptive Pairwise Critic and Asymptotic Actor"
_ICLR.cc/2023/Conference — Submitted to ICLR 2023_

### Official Review · Reviewer_6A7k · 2022-10-21

**Confidence:** 4
**Correctness:** 2
**Technical Novelty And Significance:** 2
**Empirical Novelty And Significance:** 2
**Recommendation:** 3

**Clarity, Quality, Novelty And Reproducibility:**

The paper is written clearly, although more explanations about the proposed method would be useful.
The work is rather incremental on top of SAC. At the same time, the results are not thorough and strong enough to make the work very impactful.
The algorithm is described in detail (partly in the appendix) and should be easy to reproduce.

**Strength And Weaknesses:**

The paper is written clearly. However, the organization of the paper does not convince me. A lot of space is used to describe the base algorithm SAC and very little space to describe the suggested method. For example, equation 7 and especially equation 15 describing the soft update seem unnecessary to state in the paper. Instead, I suggest the authors include at least a short description of the function modeling the state-dependent factors in the main paper and not just in the appendix. Further, Figure 1 showing the mujoco environments is not needed in the main paper and Figures 2 and 3 could be made such that they use a lot less space by plotting the legend only once and rearranging the plots in one or two rows. This would allow the experiment about the value estimates to be in the main paper.

Conceptually, I do not understand and also can not see a given argument why updating the parameters of the state-dependent networks $\lambda$ and $\mu$ according to eq. 11(i.e. the actor loss)  makes sense.

Evaluation:
The evaluation is poor. There is no reason why the proposed method APAA is compared with several baselines in Figure 2 and an extra Figure is used to compare it with SAC-t (SAC with automatic entropy tuning). This could and should all be in one Figure. Then suddenly HalfCheetah is not showing in the second Figure and judging from the performance of APAA on HalfCheetah in Figure 2 this is because it is far worse than SAC with entropy tuning. Further, the baselines for HalfCheetah in Figure 2) have clearly worse performance than what is otherwise reported in the literature.
I do not see justifications for the claim that APAA has an 'overwhelmed advantage [...] over other baselines'.

Related work: There are some papers coming to my mind that are relevant but not mentioned. In [1] a parameter Is learned to penalize overestimation of the critics. [2] adapts a parameter online during training that balances over-and understimation of the critics. Other works use a (stochastic) weighting of critics to generate better value estimates [3, 4].


[1] Cetin, Edoardo, and Oya Celiktutan. "Learning pessimism for robust and efficient off-policy reinforcement learning." arXiv preprint arXiv:2110.03375 (2021).

[2] Dorka, Nicolai, Joschka Boedecker, and Wolfram Burgard. "Adaptively Calibrated Critic Estimates for Deep Reinforcement Learning." Deep RL Workshop NeurIPS 2021. 2021.

[3] P. Lv, X. Wang, Y. Cheng, and Z. Duan. Stochastic double deep q-network. IEEE Access, 7:
79446–79454, 2019. doi: 10.1109/ACCESS.2019.2922706.

[4] Zongzhang Zhang, Zhiyuan Pan, and Mykel J. Kochenderfer. Weighted double q-learning. In Proceedings of the 26th International Joint Conference on Artificial Intelligence, IJCAI’17, pages 3455–3461. AAAI Press, 2017

**Summary Of The Paper:**

The paper studies maximum entropy reinforcement learning in an actor-critic setting. Two improvements are proposed. First, a learnable state-dependent weighting between two critics is intended to balance over- and underestimation bias of the value estimates provided by the critic. Second, the temperature parameter of the entropy regularizing the actor is also made a learnable state-dependent variable in order to smooth the entropy policy exploration. The suggestions are incorporated into SAC and results are presented.

**Summary Of The Review:**

The paper does not convince me why the proposed method makes conceptual sense. The evaluation is not thorough and hence also does not convince me that the proposed method is very useful.

---

### Official Review · Reviewer_gnMQ · 2022-10-23

**Confidence:** 4
**Clarity, Quality, Novelty And Reproducibility:** The paper is clearly written.
**Correctness:** 3
**Technical Novelty And Significance:** 2
**Empirical Novelty And Significance:** 2
**Recommendation:** 5

**Strength And Weaknesses:**

Pros: (1) The paper writing is clear.

(2) The experiments show the benefit of the proposed algorithm in several tasks.

Cons: (1) The main concern I have is why a per-state adaptive variable/weight benefits training. For example, for the state-dependent adaptive random weight that assigns weights to the two critics, is it actually mitigating the overestimation or something else?

(2) The introduction mentioned bootstrapping error, extrapolation error, overestimation error, etc. I'm confused about how the proposed method addressed them. Analysis that supports the error-mitigation claim is lacking.

(3) I'm also interested in which of the above errors are the contributing factors in policy degradation and which source of error the method addresses. I don't think a single design can address all of them at once since they are clearly not the same type of error.

**Summary Of The Paper:**

This paper proposes to use adaptive pairwise critics and adaptive asymptotic maximum entropy to address the estimation errors.

**Summary Of The Review:**

A deep and thorough analysis to support the authors' claims is lacking.

---

### Official Review · Reviewer_9zpF · 2022-10-24

**Confidence:** 5
**Clarity, Quality, Novelty And Reproducibility:** 1. The joint Q-value function is give…
**Correctness:** 2
**Technical Novelty And Significance:** 2
**Empirical Novelty And Significance:** 2
**Recommendation:** 3

**Strength And Weaknesses:**

Strength
1. The authors propose a state-dependent coefficient of the entropy regularization term that can be updated during learning. The update rule is different from SAC-v2.
2. To reduce the estimation bias, multiple Q-value estimates are often used. The proposed method adaptively mixes two Q functions, where mixing weights are state-dependent and learnable.

Weakness
1. The proposed method does not reduce the estimation bias explicitly. Specifically, the estimation bias was not evaluated in the experiments, although it is one of the main paper's motivations.
2. The state-dependent coefficient of the entropy term is not explained well. In particular, it is unclear why it is not given as a simple function \alpha(s).


**Summary Of The Paper:**

This paper deals with two issues in deep reinforcement learning. One is to reduce the estimation bias by learning a state-dependent weighting function for two Q functions. The other is to tune a state-dependent coefficient of the entropy regularization term of the policy. The state-dependent weighting and coefficient functions are updated as part of the policy parameter. The proposed method is evaluated on the standard MuJoCo benchmarks, and the experimental results show that the proposed method outperforms the baselines.

**Summary Of The Review:**

Although this paper proposes a new idea, the experimental results do not support the main claim. Therefore, the ablation studies discussed above are needed to understand how the proposed method improves performance.

---

### Official Review · Reviewer_djw4 · 2022-10-24

**Confidence:** 4
**Correctness:** 2
**Technical Novelty And Significance:** 3
**Empirical Novelty And Significance:** 2
**Recommendation:** 3

**Clarity, Quality, Novelty And Reproducibility:**

- Clarity: The method is not too hard to understand, but some of the writing can be improved for better clarity
- Quality: The overall quality can be improved
- Novelty: Not a big change from existing methods, but the adaptive part can be seen as somewhat novel
- Reproducibility: technical details, a hyperparameter table are provided to help reproducibility.

**Strength And Weaknesses:**

Strength:
- technical details: 10 seeds for each experiment, and hyperparameter table in the appendix, it seems in most cases the same hyperparameter setting is used, which is good and makes results more reliable.
- Novelty: adaptive bias control can be an interesting and somewhat novel method
- Results: compared to baselines are somewhat good assume they are reliable
- Simplicity: method is not too complicated

Weaknesses:

**Technical details**
- With the additional parameters and the extra complexity, how does your method compare to others in terms of computation and parameter efficiency?
- Although your adaptive parts are learned, there are some new hyperparameters, how hard it is to tune them?
- You are combining two main modifications, might want to ablate them

**Reliability of results and fairness of comparison**
- Why does SAC get 2000 in HalfCheetah? SAC is a very popular baseline and should get at least about 12K score at 1e6.
- For Figure 3, it seems a target entropy of 0.5 is chosen for SAC adaptive? Wouldn’t it reduce the performance of SAC adaptive if you arbitrarily set a temperature? What makes that a fair comparison?

**related work**
- “How to design a mixture of Q-values is still largely left untouched. ” not exactly sure if I agree with that. Figuring out how how get more accurate Q values is certainly important, but there are a number of other recent works that deal with this, in the paper there is no discussion on how your work relate to or has an advantage over these newer methods. (though I do acknowledge these works do not study adaptive bias control methods)
- Randomized Ensembled Double Q-Learning: Learning Fast Without a Model (ICLR 2021)
- Controlling Overestimation Bias with Truncated Mixture of Continuous Distributional Quantile Critics (ICML 2020)
- Dropout Q-Functions for Doubly Efficient Reinforcement Learning (ICLR 2022)

**soundness of arguments**
- It is unclear to me how the stability issue is evaluated. And why is there a tradeoff between efficiency and stability? Shouldn’t a more stable algorithm be also more efficient? The discussion on this point seems to be a bit vague, and there is no quantitative analysis on the results, for example, if it is about stable performance across seeds, you might want to measure that and show your method actually works better (has a better measured value) than other methods. If it is about more stable performance over test runs for a single training seed, you want to measure sth else.

**writing and clarity**
- Some minor typos or citation formatting issues, for example:
- “the twin delayed deep deterministic policy gradient (TD3) Fujimoto et al. (2018) for example”
“Several of recent works deal with errors like bootstrapping error caused by out-of-distribution (OOD) actions Kumar et al. (2019; 2020)”
- Some parts of writing is not clear for example “In Hopper environment, since the converged value is far lower than other benchmarks, the tolerance for the fluctuation around convergence is much lower, which causes…” would be good to go through the paper carefully and rewrite some of the arguments for better clarity.
- and there is a number of other issues please go through your paper and fix these writing issues.

**novelty**
- not exactly sure how novel the adaptive entropy part is... SAC adaptive already has some adaptiveness there

Other questions:
- is beta your only new hyperparameter? If so you might want to emphasize it


**Summary Of The Paper:**

The paper proposes a new algorithm over a baseline like SAC with 2 main modifications: a) adaptively learn a weight that balances between 2 critics and controls how the Q target value is computed b) adaptively learn a temperature for max entropy exploration.

The authors claim the proposed method can better balance the tradeoff between efficiency and stability. The proposed method is evaluated on 5 gym tasks and compared to a number of other baselines to show that the proposed method has better performance. Some theoretical results and some empirical analysis are provided.

**Summary Of The Review:**

The proposed method can be seen as somewhat novel and interesting, however, I have a lot of concerns on writing clarity, and on reliability of the comparison results. Additionally, the paper can benefit from more ablations, hyperparameter sensitivity study, and the claim on tradeoff should be backup up with more quantitative comparisons.

A bit difficult to recommend the paper at its current state.

---

### Decision · Program_Chairs · 2023-01-20

**Decision:**

Reject

**Justification For Why Not Higher Score:**

Unanimous agreement among reviewers. No author rebuttal.

**Justification For Why Not Lower Score:**

N/A

**Metareview: Summary, Strengths And Weaknesses:**

The reviewers unanimously agree that this paper is not ready for publication. Therefore I recommend rejection.